# Supervision, Moral Distress and Moral Injury Within Palliative Care—A Qualitative Study

**DOI:** 10.3390/ijerph22071156

**Published:** 2025-07-21

**Authors:** Pia Geuenich, Lena Schlömer, Sonja Owusu-Boakye, Henrikje Stanze

**Affiliations:** Faculty of Social Sciences, Centre for Nursing Research and Counselling, University of Applied Sciences Bremen, Am Brill 2–4, 28195 Bremen, Germany; geuenich.pia@gmail.com (P.G.); lena_schloemer@web.de (L.S.); sonja.owusu-boakye@hs-bremen.de (S.O.-B.)

**Keywords:** moral distress, moral injury, supervision, palliative care, hospice care, nurse

## Abstract

The number of people requiring palliative care is increasing. This can result in moral and ethical conflicts that may lead to psychological distress and moral injury. (MI). Solutions are needed to counteract career abandonment—supervision (SV) could be one solution. This study examines the extent to which palliative care nurses link MI to their everyday experiences and whether SV can contribute to the identification and prevention of moral distress and MI. In addition, factors that influence the implementation of, participation in, and perception of SV are analyzed. A qualitative study design was chosen for the investigation, consisting of guided interviews, narrative-generating questions with seven participants working in palliative care, and participant observation with audio recording during two supervisions of two palliative care teams with 16 participants in total. The data was analyzed using qualitative content analysis according to Mayring. The results show differences in workload between acute and palliative care wards. Time pressure and hierarchical structures promote distress and MI and are particularly evident on acute wards. The interviewees described specific experiences of MI. In addition, factors were identified that influence participation in SV. The analysis of SV showed that workload is a key cause of moral distress and can have an impact on health. SV can be used for sensitization and exchange. Implementation and acceptance of SV depend on individual and structural factors. In palliative care, signs of moral distress are present and are specifically addressed in SV, which means that SV can be a tool for dealing with stressful situations.

## 1. Theoretical Background/Introduction

Every year, at least 250,000 patients requiring palliative care receive inpatient treatment in German hospitals [1]. The increase in the number of palliative care units and the expansion of outpatient palliative care provision point to the growing need for palliative medical care. According to the German Hospice and Palliative Care Association (Deutscher Hospiz- und Palliativ Verband, DHPV), more than 120,000 professionals and volunteers are involved in the care of seriously ill and dying people in Germany. Moral and ethical conflicts can arise in this collaboration, which can place a psychological and moral burden on the professionals and volunteers and may result in the phenomenon of moral injury (MI). Close and constant communication and cooperation with each other are prerequisites for the successful implementation of the objectives and tasks that the World Health Organisation (2002) has defined as “palliative care” [2,3]. Supervision (SV) could offer a good opportunity to promote communication and cooperation.

In addition to their professional duties towards patients, palliative care nurses are often exposed to dealing with negative emotions, the communication of bad news, the lack of prospects for recovery, repeated confrontations with death and grief, as well as the suffering of patients. The increase in complexity in care situations, such as voluntary stop eating and drinking and the issue of assisted suicide, also presents palliative care with new ethical challenges and responsibilities that have thus far been given little or no consideration in professional or institutional policy in Germany [4]. At the same time, precarious working conditions in relation to ethical guidelines, such as the objectives and tasks of the WHO, the ICN Code of Ethics for Nurses, and personal moral attitudes and ideas, act like a magnifier in ethical conflict situations [5]. This problem can lead to MI among nurses, a phenomenon defined as a psychological, biological, spiritual, behavioral, and social impact caused by an event that is performed or observed and cannot be prevented. It is deeply ingrained and violates deeply held moral beliefs and expectations, such as triage, hygiene measures, or social isolation [6,7]. It is preceded by a previous experience of moral distress, which is a situational problem caused by the circumstances in which a person finds themselves [8].

For several years, researchers from the USA have been investigating the topic of MI in the healthcare system there. Studies have shown that nurses affected by the social phenomenon of MI are more likely to suffer from anxiety disorders, exhaustion, depression, and other mental illnesses [9,10]. During the survey period, there were no significant studies in the field of MI in German-speaking countries, which is why literature from Anglo-American countries was primarily used.

Moral Distress (MD) and Moral Injury (MI) are distinct forms of moral suffering experienced by healthcare professionals, particularly in palliative care, when confronting ethically challenging situations. MD arises when individuals are constrained from acting according to their moral convictions, while MI reflects the profound psychological impact of actions that violate personal moral beliefs. In contrast to burnout syndrome, which is characterized by emotional exhaustion and depersonalization, MD and MI are rooted in ethical conflicts and moral dissonance [11].

In order to counter the phenomenon of MI and to take action against career flight as a further consequence, practical solutions must be found.

In order to ensure the health of employees and team cohesion, it is important that institutional responsibility is assumed and preventive measures are put in place [5,12].

The institutional implementation of SV leads to an “improvement in communication in the workplace and promotes cooperation in teams” and is already being practiced by palliative and psychiatric healthcare teams in Germany [13]. SV makes it possible to discuss and reflect on personal experiences, concerns, and professional conflicts and to find concrete solutions [13]. The ethical guidelines of the German Association for Supervision and Coaching (Deutsche Gesellschaft für Supervision und Coaching, DGSv) define supervision and coaching as “professional forms of counselling that serve to maintain, expand or restore the capacity of individuals and/or organizational units to perform […]. On the basis of a work agreement, supervisors and coaches advise professionals on their professional behavior, their role design, their working relationships and cooperation with each other and with their clients […]” [3,4]. According to a study from England, the willingness of professional careers in palliative care to participate in SV and the perception of it depend on several factors [9]. While SV has been examined in Germany, possible factors that influence the successful implementation, willingness to participate, perception of SV, or the detection of possible signs of moral distress in SV did not form the basis of such studies [8,10].

✓In this context, as part of the study project Supervision and Moral Injury in Palliative Care (SEMPF), the project team took a closer look at the topic of MI and SV in palliative care, focusing on the following research questions:✓How do palliative care nurses perceive the workload?✓Can nurses identify themselves with the term MI, and have they experienced situations relating to this topic?✓Do nurses think that SV can be a method to identify and prevent the phenomenon of MI?✓What factors influence the successful implementation, willingness to participate, and perception of SV in palliative care?✓What signs of moral distress can be recognized in SV?

## 2. Methodology

This project took place from April 2023 to December 2023. The recruitment for both inquiries took place from May 2023 to August 2023 via the digital distribution of flyers by the German Society for Palliative Medicine (Fachgesellschaft der Deutschen Gesellschaft für Palliativmedizin) and by Alpha Rheinland, contact point in the state of North Rhine-Westphalia for palliative care, hospice work and support for relatives, as well as via the cooperation partners of the City University of Applied Sciences Bremen. Due to the parallel analysis using inductive category development, data saturation was achieved after seven interviews and two participant observations, leading to the termination of further participant recruitment.

### 2.1. Participants

Seven participants from diverse areas within palliative care took part in the individual interviews, providing a multifaceted perspective on the emotional, ethical, and structural challenges inherent to their field. The sample included professionals from both inpatient and outpatient settings, reflecting a broad spectrum of experiences and working conditions. This diversity allowed for a differentiated understanding of how factors such as team culture, care structures, and patient characteristics shape the perception of workload, moral stress, and professional identity in palliative care. Their insights contributed significantly to identifying moral injury as a relevant phenomenon and highlighted the role of supervision as a supportive mechanism across different care environments. Two palliative care teams, comprising a total of sixteen participants, were involved in the participant observation component of the study. One team represented an inpatient setting with five members, while the other operated in an outpatient context with eleven members (Table 1).

### 2.2. Interviews (Palliative Care Nurses)

Guided interviews with narrative-generating components were conducted to determine the workload of nursing staff in inpatient and outpatient palliative care. The question of the extent to which nurses can identify with the term *MI* and have experienced situations involving it was also investigated [14]. Only nurses from the field of palliative, care were approached for the interviews. This ensured that the interviewees were familiar with the SV method and nursing care tasks. This also allowed a comparison between the workload in palliative care and the workload in the general nursing setting, with all facets of basic care and treatment. The interviewees had the opportunity to conduct the interviews in person or online via video calls, and all chose the digital format. Given the impact of the COVID-19 pandemic, the research on moral injury in the nursing context—therefore, the increased workload and ethical strain experienced during this period—was also taken into account in the interviews [7]. In the Appendix A, the interview guide with the narrative-generated questions is attached.

### 2.3. Participant Observation (Supervision in the Area of Palliative Care)

The participant observation, which was audio-recorded in parallel, was used to determine (1) the factors influencing the willingness to participate and the assessment of successful implementation of an SV, (2) the perception of SV as a concept for improving the (moral) workload, (3) SV as a way of assessing moral distress among the participants. Due to the poor response, several palliative care teams were subsequently contacted directly by email. One project member was in the room during each observed supervision session. Apart from the presence of the project member, there was no active intervention in the course or content of the supervision.

### 2.4. Analysis of Individual Interviews and Participant Observation

The data collected in both studies were analyzed using the inductive approach of Mayring’s qualitative content analysis; the computer-assisted MAXQDA 2022 programme was used for the analysis [14,15].

Mayring’s qualitative Content Analysis was employed in the project as a systematic, rule-guided approach to analyze textual data. This method allows for the structured condensation of complex information and the inductive development of categories. It ensures closeness to the original material while facilitating the recognition of overarching themes and patterns. Furthermore, the approach enhances the intersubjective traceability of the findings, thereby strengthening the scientific validity of the study [15].

Mayring’s qualitative content analysis is grounded in a structured and methodical framework for interpreting textual or visual data. Central to this approach is the formulation of well-defined research questions, the development of a coding scheme informed by theoretical constructs or existing empirical knowledge, and the systematic application of these codes to the data to ensure consistency and reliability. Furthermore, Mayring underscores the analytical value of paraphrasing as a means to generate intelligible and meaningful categories. Equally essential is the transparent and comprehensive documentation of each step in the analytical process to enhance traceability and methodological rigor. To reduce potential subjectivity, research workshops were held regularly (every four weeks during the coding phase) to discuss interview passages and legal codes (especially at the higher level of abstraction, as envisaged by Mayring). The research workshops were attended by individuals from nursing science and nursing education (HS, PG), sociology (SOW), and nursing practice (PG, LS).

In both qualitative surveys, personal data was collected and secured in accordance with the guidelines of the General Data Protection Regulation (GDPR) [16]. In order to comply with the requirements of the GDPR, a project cover letter and a declaration of consent were drawn up to ensure that the participants were informed about the protection of their data. The declarations of consent had to be signed before the start of the study. The participants’ data were pseudonymized.

### 2.5. Ethical Considerations

Following an ethical review, the interprofessional research ethics team of the City University of Applied Sciences Bremen did not raise any ethical concerns (No. ISPF0102401). Voluntarily participating nursing staff and healthcare staff in supervision received written information about the study and their participation in an interview or observation of their supervision. In this invitation, it was pointed out that the data would be pseudonymized and that identification by third parties is not possible in any publication. It was explained that participation was voluntary and that participants could also withdraw from participation at any time without giving reasons. Participants of each part of the study gave written informed consent.

## 3. Results

Seven participants were recruited for the individual interviews; the results and corresponding main categories are presented first.

Two teams were recruited for the participant observation of the SV. One team was represented by four participants from inpatient palliative care, the other team from outpatient palliative care had twelve participants. The teams studied were declared suitable for the project if they focused on palliative care and conducted regular SV sessions. These results follow on from those of the interviews.

Two category systems were developed from the collected data in analysis steps according to qualitative content analysis, the main categories of which are presented below based on the thematic focus and forms of data collection (Figure 1).

### 3.1. Results of the Individual Interviews

#### 3.1.1. Experiencing the Workload in Palliative Care

The analysis of the interviews showed that the workload and the physical and psychological stress are very subjective. Each nurse perceives this differently due to their personal coping mechanisms, which are reflected in their own feelings of stress. Based on their professional experience, nurses were able to draw a comparison between the workload on a hospital ward in other care settings and that in palliative care. The work in acute inpatient care with (mostly) curative approaches was described as very time-consuming, with prolonged workdays (extremely long working days). The perceived workload due to staff shortages and increased patient numbers is extremely high. These high levels of stress meant that important tasks could not be completed and led to negative feelings of stress (distress).

Compared to various other areas in the hospital, the workload in the field of palliative care is described as “completely different”.

“I then moved to the hospital [.], the workload was once again of course extremely high, it is said that you are usually supposed to look after ten patients on average as a specialist and there were usually twice as many, if not almost three times as many. And then it was the case that at some point you couldn’t manage certain simple but very important things [...], or not in a timely enough fashion, and at some point, you no longer left everything professionally behand at work, but instead dragged it home with you and asked yourself afterwards, did I do that or not?”. (MI_001)

Due to a higher staffing ratio and better interaction with each other, the formal workload in outpatient and inpatient palliative care seems to be reduced for carers and therefore more pleasant and easier to implement. Nevertheless, the interviewees experience a burden and strain that can only be recognized differently compared to the burden on a regular hospital ward.

“It is different now on hospice duty. My workload is less than in hospital [...] It starts with the way we deal with each other. We treat each other differently [...] It is altogether different from the hospital. I always have someone to talk to here if I have any things I want to discuss”. (MI_004)

“I think the clientele is changing in the palliative care sector. The sick people who come to us. The [...] constantly new and difficult symptom management is also changing. Also, what we often pass on, of course, that there is no idealized dying, that is changing a lot right now and naturally in the course of the social changes, the political changes, we are also noticing a change in our context, it is a high workload because the people who come to us are sicker than they were years ago”.(MI_005)

The interviewees’ statements show that the cause of their workload can apparently be attributed primarily to the current structures in the healthcare system and not just to the difference between the areas of acute inpatient and palliative care. Furthermore, it can be seen that if nurses had more time and more staff available, there would be no need to prioritize activities at the expense of the quality of patient care. In addition, the distribution of tasks could be more clearly defined and, according to the interviewees, relieve the burden on carers. It is also clear that the ideas of living and dying are facing a social reinterpretation, concerning what a “good death” might mean. However, this rethinking in practice also requires more dialogue on ethical issues, both between professional groups and with relatives and loved ones.

#### 3.1.2. Identification with the Phenomenon of Moral Injury

MI is a consequence of actions taken that are contrary to one’s own morals, values, and norms.

The analysis shows that the actions and decisions of other professional groups can apparently influence the work of nursing staff. This can include, for example, decisions made by doctors that are not supported by the nursing profession in the context of palliative care and therefore result in dissent. Actions within the nursing team can also go against the morals and values of individual carers.

According to the interviewees, such actions could cause their own moral compass to waver and thus create the source from which MI can arise.

“that you always have the feeling that you could have done more, that you should have done more and [...] the person was probably not receiving the best possible care at that moment and that would also create a moral dilemma for me”.(MI_002)

“of course, we also experience, and we have experienced this more frequently recently, new case decisions again and again, borderline decisions that we have to support, but as carers they influence us a lot, in our moral compass, I would say, because the medical team is always looking more and more at therapies, therapy goals or looking for more support or more possibilities to offer the sick person something, to make an offer”.(MI_005)

With regard to situations already experienced by the interviewees that violated their own moral concepts, situations were described as stressful. In these difficult circumstances, the interviewees described simply functioning and yet feeling torn inside.

“what I and my colleagues talked about and also suffered a lot from was not being allowed to have visitors [note: during COVID measures] visit patients and somehow having to orientate themselves under these constantly changing clear guidelines or something. It was really unbearable for a while to only be allowed to say goodbye and then come in individually”.(MI_003)

#### 3.1.3. Identification and Prevention with the Help of Supervision

Communication is obviously an important factor for the interviewees in their work in palliative care. Discussions about stressful situations serve to provide relief to them and were mentioned as a way of identifying the phenomenon of MI. In general, there is a desire for more communication within a team in order to discuss stressful situations and thus give colleagues the opportunity to relieve stress.

“[…] on the one hand, it is important that there are people who are already aware or have knowledge of this phenomenon and [...] I think it is very, very important. So, both are important to me, on the one hand that there are people who know about it and who are aware of it, who can guide and lead others towards it, so to speak, and on the other hand that there is openness and communication in the palliative context”.(MI_005)

According to the interviewees, SV therefore seems to be a good tool for discussing stressful situations, and was confirmed by them as an identification opportunity for MI. In addition to discussing stressful situations and making joint decisions in difficult situations, SV apparently also serves to give the team an opportunity to discuss problems with each other, but also to strengthen team spirit and jointly develop proposals for solutions to improve daily professional practice.

“I do believe that supervision is an instrument that can achieve good results, but only if you can really assess your team. That it can cope with it. But then, if the team is ready for it, then I think it’s a very good way to create communicative spaces in which such violations can become apparent and perhaps individual nursing staff can recognize that [...]. OK, I’m not the only one who is confronted with this problem. That alone could help a lot”.(MI_002)

### 3.2. Results of the Participant Observation

#### 3.2.1. Signs of Moral Distress in Supervision

The analysis of the participant observation shows that teamwork is considered very relevant for successful collaboration in challenging palliative work. It is evident that the team must find ways to jointly overcome the challenges in a joint communication process, both within and across professions. A lack of collegial support and a reduced presence in the workplace can apparently lead to participants feeling like lone warriors in their profession and in their team, thus experiencing little community and cohesion.

“I realize myself, because I have a part-time job and am not always present, that you really are a lone warrior in palliative work [...]”.(SUPA4)

In palliative work, healthcare workers are often confronted with challenging situations and react in different ways. It becomes clear that there are situations and interactions with patients that trigger negative emotions in staff. These emotions can apparently lead to a reaction within the staff, which makes them want to escape the situation quickly—i.e., to flee. This flight reaction can even trigger physical complaints. The temporal intensity of the work and the contact with the patients obviously have an effect on the perception of one’s own effort. Having time away from the difficult situations —for example, by taking days off—should therefore have a positive effect on this effort.

“It always triggered a kind of aggression in me and also an escape reflex. I thought, I have to get out of here now […]”.(SUPA3)

“There is sometimes aggression in the countertransference and you have the feeling that you have to get out of here because you can actually feel it physically: somehow it tightens my throat […]”.(SUPA3)

During and after a long accompaniment of individual palliative care patients, it seems to be important for the participants in the participant observation to exchange information about care and treatment. This exchange makes it easier to make decisions and explore new options for action. It would appear there is not enough time for this exchange, although a more regular and longer time frame for this is desired.

“It’s probably easier to make a decision if I can talk about it and someone says I would have done the same thing or someone says, great that you did it that way, maybe next time that’s an option or whatever […]”.(SUPA1)

Working in palliative care requires professionals to make many decisions. The participants of the SV do not find the line between right and wrong easy, because this depends on the justifications and the comprehensibility of the treatment goal decisions, whereas the views and wishes of the patients must be taken into account. The decision between morally right and wrong does not automatically have to go hand in hand with one’s own view of what would be “right” in this case. For example, something can be declared as “right” for the individual and still not be implemented. According to the participants, a disagreement can therefore arise between their own idea of what is morally right and wrong and the attitude of others in the team. This dissent can also arise independently of the team members due to the philosophy of the organization in which they work.

“[…] you first have to look at what you represent as a team, i.e., what the team attitude is, of course everyone has their own personal opinion on such a topic and I can imagine that there are always differences between what you say and what you represent as a team, which may not exactly be your personal opinion, and I can imagine that there are always, I don’t know, moral violations”.(SUPB7)

Making independent decisions without consulting colleagues was named as a further aspect of the stressful experience of palliative care. In this survey, nurses stated that they felt forced to make decisions that appeared to be outside their own area of expertise and responsibility, particularly when there was no doctor present. Even when doctors are present, nursing staff in palliative care are apparently required to make decisions or suggestions despite their lack of qualifications, according to the study participants. This apparently has a negative impact on their emotional state and leads to dissatisfaction, as nurses feel internally responsible for decisions made outside their area of expertise.

“Yes, I find when I am asked by the doctor, so what dosage? For me it’s like-, then I have a thousand question marks and I think: no, something is not right here right now, because I haven’t had this training to be able to decide on this or that dosage”.(SUPA5)

The participants increasingly described the perception of not being able to provide justice to all patients or never being able to offer enough. During the interaction, the patients conveyed the feeling that the staff were not doing their job thoroughly or completely. Not being able to fully meet the needs of the patients is perceived by the participants as a “deep disappointment in life”.

“Although so many times—but it was still that I couldn’t help them. That’s actually a very deep disappointment in life, isn’t it?”(SUPA3)

#### 3.2.2. Factors Influencing Perception, Implementation, and Willingness to Participate in Supervision

Working together on problem areas seems to be something that has positive effects for the participants and results in the desire to expand this further. It is apparently important to the participants that they use the time productively and work on something tangible.

“[…] that we all worked together on these points, and in the end, it emerged that this is something that is good for us and that we want to work on […]”(SUPA3)

Participants in SV and the supervisors themselves have different expectations of the sessions. It can therefore happen that their own ideas or plans for the SV are not fulfilled. Expectations of the SV can obviously be influenced by factors such as external circumstances or team dynamics. Changing supervisors can also apparently lead to unfulfilled expectations.

Communication within the team regarding the framing and content of the SV seems to have an influence on the benefits of SV, which changes the perception of SV, which can have a positive or negative influence on the willingness to participate. A lack of agreement in this regard and therefore a lack of information being passed on about the past content of an SV leads to discomfort at the meetings because people gather the feeling that they can neither follow nor contribute appropriately.

“What was actually discussed? (laughter). But you can kind of tell from that what’s on top, even at work, just that we’re very busy with our work and I really wasn’t there for your introduction, at the last team meeting, I didn’t even realize what it was about. I also haven’t received any minutes”.(SUPA4)

Aspects that influence a person’s willingness to participate in and implement an SV are their time resources. For example, it appears that a highly compressed workload and little time for relaxation have an influence on the willingness to participate.

“And then to say, I can also understand that if you’ve worked, say 7 days or 7 nights, then of course you’re exhausted on Tuesday and say O.K. now I have to find my rhythm again, then I’m not really super willing to participate here—it’s always such an issue.”(SUPA3)

The perception of the effectiveness of supervision has an impact on the willingness to participate in supervision. This perception is strongly influenced by the supervisors. It seems to be the case when there is a high turnover of supervisors in a team, and the participants miss professional continuity in the guidance. The systematic procedures or methods of the supervisors seem to influence whether the SV is perceived positively. In contrast to the systematic approach of some, other supervisors do not seem to have a fixed approach and give the participants more freedom of direction. Depending on the type, this can have a positive or negative influence on the perception of the SV.

“So, since 2007, we have had, I don’t know, five, six, you have to correct me, seven, eight different supervisions, some of whom we had for years, some of whom only for a short time”.(SUPA3)

“You get to know the spectrum of what everyone offers and it’s always personalized and very different, depending on the person who introduces it; the supervisor [...]”(SUPA3)

It is obviously important for the participants in the participant observation to be able to talk about stressful situations that do not find space in everyday work and short team meetings. Here, the SV seems to offer a good opportunity to reflect on one’s own actions and decisions with the team and the support of supervisors. According to the participants, the focus of the SV should be on what benefits the team can derive from it, according to which the style of guidance by the supervisors in the SV should be based.

“But then it’s not just something for this moment when such situations occur, but it’s also for others who might experience this again, or who have already experienced it, or something, then we can all benefit from it and discuss what we would have done or something, so yes, I think it’s good.”(SUPA2)

## 4. Discussion

The SEMPF project focused on the thematic priorities of MI and SV.

The aim of analyzing MI was to ascertain the current workload of palliative care nurses, based on individual interviews, and to investigate the extent to which they can identify with the phenomenon of MI.

The interviewees cited understaffing and the resulting extra work as the cause of their workload. This, in turn, results in a lack of time to provide patients with the quality of care that they deserve from the point of view of the nursing staff interviewed. The (state) regulations and laws under which nurses had to work during the COVID-19 pandemic were also emphasized as highly stressful [17]. The findings of this survey are in line with those of the study by Happel et al., who conducted a qualitative survey on the causes of stress among nurses in 2013. The consistent stressors were as follows: high workload, lack of doctors, insufficient support from management, relatives of patients, e.g., through their demands on the treating professional groups [18].

Performing actions that go against one’s own moral and ethical attitudes are among the causes of MI, i.e., moral injury. The interviewees could either identify with the phenomenon and had already experienced situations of this kind themselves, or could at least understand example situations, even if they had not consciously encountered this phenomenon before. Nevertheless, it became clear that the workload to which nurses are exposed in palliative care sometimes contradicts ethical and moral attitudes and represents stressors that can lead to MI. This study only surveyed the workload of palliative care nurses and their possible identification with the phenomenon of MI; it did not explicitly address the individual stress factors. Nevertheless, some of the interviewees addressed physical and psychological stresses and reported an impact on physical resilience and mental health. These statements are in line with the findings of Jack & Kotronoulas (2023) and those of Thibodeau et al. (2023), who analyzed the possible links between moral injury and factors of occupational wellbeing and mental health in their reviews [19,20]. In particular, the individual emotional and spiritual needs of healthcare professionals were largely ignored [21].

These insights are further supported by Clayton and Marczak (2022), who conducted a thematic synthesis of stress, anxiety, and burnout among palliative care nurses. Their analysis identified three dominant themes: “When work becomes personal,” “The burden on mind and body,” and “Finding meaning and connection” [6]. These findings underline how the emotional proximity to dying patients, coupled with systemic challenges such as chronic understaffing and inadequate institutional support, creates not only professional strain but also deep personal and ethical conflict. This overlap between personal and professional boundaries can significantly contribute to the emergence of moral injury, especially when core values such as dignity in care cannot be upheld due to time or resource constraints.

Moreover, the study by Amsalem et al. (2021) investigated psychiatric symptoms and moral injury among healthcare professionals during the COVID-19 era and found strong correlations between experiences of moral conflict and symptoms of depression, anxiety, and post-traumatic stress [9]. These findings suggest that situations in which healthcare professionals are forced to act in contradiction to their ethical beliefs—such as prioritizing patients under resource scarcity—significantly elevate the risk of moral injury. This highlights the urgent need for structured support systems, such as supervision or peer-group reflection, to mitigate long-term psychological harm.

To date, there are few studies outlining identification options, prevention approaches, and treatment possibilities for moral injury to counteract the phenomenon of MI and its extent.

SV could be one way of addressing the issue in a non-judgmental space. The interviewees agree that a professionally led SV is suitable for identifying moral injury. Due to the lack of scientific findings on the effect of SV in MI, research interest in this area should increase, as the few existing studies indicate that open and targeted communication about traumatic stress shows treatment and prevention success [21].

The focus of the analysis on SV was to examine factors influencing the willingness to participate, the assessment of successful implementation, as well as the perception of SV. In addition, SV was to be focused on as a way of assessing moral distress among the participants. The data was collected in participant observations with parallel audio recordings of the SV sessions.

The analysis of the data material showed that the perception and implementation of SV is linked to the supervisors. According to Puffet and Perkins (2017), palliative care nurses appreciate supervisors who are good listeners, show understanding, and are trustworthy. It is important to make it clear what the aim of the supervision is and, at best, to offer the option of group and individual supervision [22]. Mittler, Petzhold & Blumberg (2019) were able to show that supervisors should have ‘a high level of communicative, emotional and social competence’. In addition, the training, professional competence, and experience of the supervisor seem to play a decisive role [23]. However, the professional title ‘supervisor’ is not protected in Germany, such that in principle, any person could offer SV and call themselves a supervisor. Before commissioning supervisors, clients must therefore research the training and further education of the respective person in order to ensure a certain quality of supervisor [24].

The study conducted here showed that participants often do not take part in SV due to a lack of time and mental resources, even if there is a need for it. The participants refer to working several days in a row or do not give a reason for the absence of their colleagues. Peters, Heckel & Ostgathe (2020) show that there is less talk of scarce economic resources in palliative care than in other disciplines, although the scarcity of resources can also be found here [25]. Puffet & Perkins (2017) describe that participants were less likely to take part in SV if participation was considered voluntary. The obligation to participate was perceived as positive by the participants themselves, as it gave staff the capacity to participate in their working hours, thus allowing them a ‘protected time’ to participate in SV and enabling regular participation and successful implementation of SV [22]. SV for professionals and volunteers is mostly funded by employers and is highly recommended by professional associations such as the German Society for Palliative Medicine [2023b]—yet to date, no general obligation for the implementation exists. An obligation for participants to take part would lead to mandatory participation in the counselling service for the professional groups involved in palliative care, but at the same time would also give them the secure time to participate on behalf of the institution [26,27]. Statutory funding, e.g., in the German Social Security Code V, could enable SV to serve as a primary and secondary preventive measure to avoid the symptomatic effects of moral distress and MI by dealing with morally stressful work situations.

The analysis showed that the participants make decisions in their day-to-day work that they do not attribute to their profession. It became particularly clear that professional carers find themselves in situations in which they are required to make decisions that they themselves see as the responsibility of physicians. This includes, for example, the selection and dosage of medication. In Germany, nurses are currently not allowed to prescribe medication; this authority lies with doctors alone. Doctors are allowed to delegate measures to nurses that would otherwise be medical tasks, but nurses can refuse the task if they do not feel able to carry them out themselves. These delegated tasks can be very far-reaching, and the service is then mainly billed to the health insurance company by the doctors [27]. This is one possible reason why nurses do not identify with certain tasks in Germany.

Interprofessional teamwork on an equal footing is necessary for the support and treatment of people with palliative care needs, whereby specific tasks can no longer be carried out by just one professional group, but can also be performed by other professions, although the tasks are limited by professional regulations [28,29]. In palliative care, the tasks of doctors are usually clearly defined, whereas the tasks of other professional groups are not defined according to Goudinoudis (2018) [30]. A meta-analysis from Norway also describes that hardly anything is known about the role of professional careers [31]. The clear structuring of interprofessional care could resolve or at least support the uncertainties regarding the distribution of tasks and responsibilities. A palliative care delegate also organizes further education and training measures for in-house employees and could thus initiate the development of basic skills in palliative care [32]. The German Society for Supervision and Coaching (2017) counts the clarification of precisely described “roles and responsibilities in the working world” [33] among the tasks of supervisors. Participation in SV could therefore help to clarify the distribution of roles and responsibilities for all employees and reduce misunderstandings in everyday working life.

The study conducted here was able to show that it is particularly important for participants of the SV to be able to exchange ideas with their colleagues. Despite the need for dialogue expressed by the participants, this is often not feasible in everyday working life. More structured opportunities for exchange—such as regular team meetings—are often discouraged by higher hierarchical levels. In the study by Puffet & Perkins (2017), this exchange is described as “informal support in the team”. This is an informal exchange between colleagues from the same profession or at an interprofessional level [22,33]. This type of exchange is suitable for a debriefing immediately following a stressful situation, while the SV is intended for in-depth reflection on the event. Informal support in the team would also be an important tool for dealing with stressful situations and can be seen as a team-building measure [22]. Encouraging employees to engage in informal and collegial dialogue could therefore strengthen the sense of community and cohesion in the team and, at the same time, have a positive influence on how individuals deal with stressful situations. A practice-oriented concept should be developed for the regular and low-threshold use of this measure in order to be able to guarantee this during everyday working life [34].

An additional significant finding concerns the interplay between structural, professional, and individual factors that influence both the successful implementation of supervision (SV) and the willingness of staff in palliative care to participate. According to the German Hospice and Palliative Care Association (Deutscher Hospiz- und Palliativ Verband, DHPV) (2022), the high emotional demands and complexity of work in palliative care require structured support mechanisms, such as SV, to be sustainably integrated into daily routines [1]. However, this integration often faces challenges due to time constraints, lack of institutional prioritization, and insufficient interdisciplinary coordination [3,34]. The success of SV implementation depends greatly on transparent communication of its objectives, a supportive organizational culture, and the inclusion of all professional groups within the multiprofessional team [2,3]. Importantly, the willingness to participate in SV is enhanced when staff perceive it as a protected and meaningful component of their professional development and psychological wellbeing [1,2]. In the context of SV sessions, typical signs of moral distress can be recognized in the form of recurring expressions of helplessness, ethical ambivalence in decision-making processes, and reports of perceived responsibility without corresponding authority, particularly from nursing staff who feel caught between professional duty and institutional constraints [4,5]. Klotz et al. (2023) underline that such manifestations of moral burden highlight the urgent need for professional role clarity and systemic support structures, which SV can effectively address when properly implemented [5].

## 5. Limitations

The SEMPF study is a qualitative study with an explorative approach. It fulfils the quality criteria of qualitative research and provides initial theoretical content on the phenomenon of MI and the significance of SV [33]. The research team for this study consisted of people with a professional nursing background, which meant that care had to be taken to maintain a distance from the field when analyzing the collected data in order to avoid bias. In order to ensure this, the data was randomly analyzed together within a research workshop of the project team with professionals from sociology, nursing science—who had never or no longer worked in nursing—and practicing nursing.

A limitation of this study is that although this study intended to cover the relevance of gender specific differences, only one man agreed to participate in the interviews; the rest of the participants were women. The reason why many men did not participate is not clear. Further research is necessary to deepen these aspects, especially from the perspective of men who may have different views on situations. Covering only seven interviews and two supervisions in total, the sample was nevertheless sufficient to reach data saturation.

## 6. Conclusions

The findings of this study underscore the complex interplay between workload, moral distress, and the supportive role of supervision in palliative care. While the overall workload in palliative care is perceived as more manageable than in acute inpatient settings, the emotional and ethical burden remains significant due to the complexity of patient cases and the frequent exposure to End-of-life decisions. Nurses described moral injury (MI) as arising when actions, often shaped by external structures or interprofessional dynamics, contradict their personal or professional values. Situations such as insufficient time for care, ethical conflicts with medical decisions, and the emotional toll of limited patient autonomy—particularly during the COVID-19 pandemic—emerged as central to this phenomenon.

Supervision (SV) was identified as a critical tool for recognizing, processing, and potentially preventing moral injury. It creates a structured space for reflection, emotional relief, and ethical discourse, thus strengthening team cohesion and individual resilience. However, its effectiveness depends heavily on consistent facilitation, clear communication, and team readiness. Barriers such as high workload, inconsistent participation, and frequent turnover of supervisors can undermine the perceived value of SV and reduce willingness to engage.

Overall, the study highlights the need for a stable and well-integrated supervision culture in palliative care settings to mitigate the risks of moral injury, support professional identity, and enhance ethical decision-making in the face of increasing care complexity.

## Figures and Tables

**Figure 1 ijerph-22-01156-f001:**
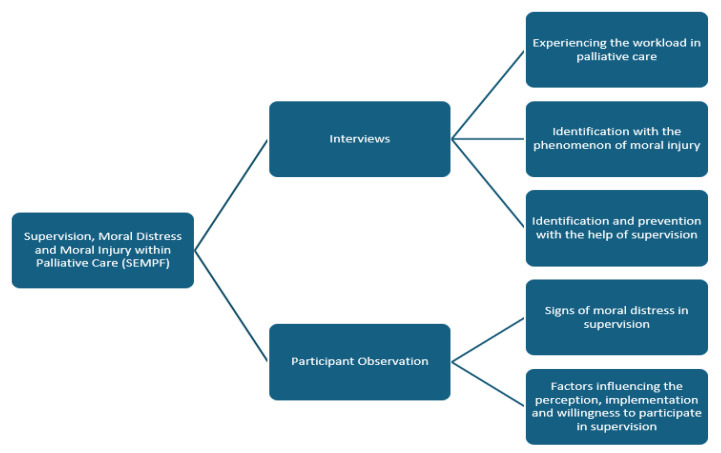
Categories Qualitative Analysis.

**Table 1 ijerph-22-01156-t001:** Characteristics of Participants.

	Number of Participants (MI)	Number of Participants (SV)
**Sample size**	7	16
**Gender**		
male	1	3
female	6	14
**Age**		
25–45 years	3	12
46–64 years	4	4
**Workplace**		
Hospice	1	0
Inpatient care	2	5
Outpatient care (adults)	2	11
Outpatient care (children)	2	
**Years of** **Service in** **Palliative** **Care**		
3–10 years	4	13
11–20 years	3	3

## Data Availability

The participants of this study did not give written consent for their data to be shared publicly, so due to the sensitive nature of the research supporting data is not available.

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
