# Peer review of "Supervision, Moral Distress and Moral Injury Within Palliative Care—A Qualitative Study"

_ijerph, 2025, doi:10.3390/ijerph22071156_

Round 1
Reviewer 1 Report
Comments and Suggestions for Authors
Thank you for sending me your article, "Supervision and Moral Injury in Palliative Care: A Qualitative Study."
As you mention, studies on this topic have not assessed feelings of helplessness and hopelessness related to moral injury caused by traumatic events in the context of hospital care. This concept can be useful for understanding the experience of nurses who perform their work in sometimes extremely demanding and even traumatic situations. However, I believe some aspects need to be clarified.
In the introduction, there is no mention of the relationship or differences between moral injury and moral distress, which I consider important to clarify. This could also apply to other terms: burnout, stress of conscience, regret over an ethical situation, ethical distress, guilt without guilt, and existential suffering with the infliction of pain.
The objective of the article is not clear to me, although the discussion mentions determining the current workload of palliative care nurses, based on individual interviews, and investigating their degree of identification with the MI phenomenon. This does not address all the questions raised.
How do palliative care nurses perceive their workload?
Do nurses identify with the term ME and have they experienced situations related to this topic?
Do nurses consider SV as a method to identify and prevent ME?
What factors influence the success of implementation, willingness to participate, and perception of SV in palliative care?
What signs of moral distress can be recognized in SV?
Method:
What was the period of the interviews and participant observation?
COVID-19 is referenced in the results and conclusions.
Why was the inductive approach used instead of another data analysis method? What is its advantage over others?
Strategies used to meet data quality criteria presented in the results.
Discussion:
In the discussion, it is necessary to contextualize the findings with the existing literature.
In conclusion,
I consider it appropriate, but I could broaden understanding of the term EM and propose a theoretical interpretation that differentiates it from moral distress and burnout syndrome.
Check the references, some of them do not have the same format.
There are typographical errors throughout the text, check them (e.g. Organisation, carers).
Reviewer 2 Report
Comments and Suggestions for Authors
Supervision and Moral Injury within palliative care – a qualitative study
General Comments:
The manuscript addresses an important and timely topic with high relevance both from a scientific and a societal perspective. The integration of supervision can contribute to the identification and prevention of moral injury.
Comments to the Authors:
Strengths:
- The choice of conducting a qualitative study is appropriate and valuable for an explorative study.
- The methodology is well designed, clearly presented, and rigorously applied.
- The results are analyzed and discussed in a clear, objective, and well-founded manner.
Specific Comments:
Comment 1: Title: Should add the terms moral distress
Comment 2: Abstract: should indicate the number of participants and who they are
and in which care contexts they were studied.
Comment 3: Introduction: Needs better clarification of the support of supervision, moral injury in palliative care. There is a reference 34, this is an error.
Comment 4: Methodology: You need to start this topic by identifying the qualitative methodology. Did you follow a reference author to guide application of the method? Which author?
What questions were asked? Where were the interviews carried out?
Comment 5: Explain the sampling method.
In the topic interviews and in the topic participant observation there is a repetition of a sentence that refers to the form of recruitment.
You could have a topic just about the participants, explaining the form of sampling, recruitment, inclusion and exclusion criteria.
Comment 6: Results:The first 2 paragraphs refer to the participants in the study. These could be in the topic suggested above. The characteristics of the participants could be explained here. It would be important to identify for example, length of time working in the field of palliative care, age, gender.
Comment 7: In the results you should add a diagram illustrating the organization of the categories to help understand the results. Should find a way of linking the textand the interview transcript so that the text is more harmonious and easier to read.
Comment 8: Limitations:“The research team for this study consisted of people with a professional nursing background”, indicate the measures taken so that data collection was not influenced by this factor?
“The sample was nevertheless sufficient to reach data saturation” Justify the reason for limiting the number of participants.
Comment 9 :Conclusion: It should be restructured to give more prominence to innovative results, the impact of VS in relation to IM and contributions to practice.
Conclusion:
The manuscript makes a relevant contribution to the literature, but I recommend major revisions.
Round 2
Reviewer 1 Report
Comments and Suggestions for Authors
The present topic is of considerable interest. I would like to express my gratitude for the attention you have given to the issues I previously raised during your presentation.